# Development of Osteoporosis Screening Algorithm for Population Aged 50 Years and above in Klang Valley, Malaysia

**DOI:** 10.3390/ijerph17072526

**Published:** 2020-04-07

**Authors:** Shaanthana Subramaniam, Chin-Yi Chan, Ima-Nirwana Soelaiman, Norazlina Mohamed, Norliza Muhammad, Fairus Ahmad, Pei-Yuen Ng, Nor Aini Jamil, Noorazah Abd Aziz, Kok-Yong Chin

**Affiliations:** 1Department of Pharmacology, Faculty of Medicine, Universiti Kebangsaan Malaysia, Jalan Yaacob Latif, Bandar Tun Razak, Cheras 56000, Malaysia; shaanthana_bks@hotmail.com (S.S.); chanchinyi94@gmail.com (C.-Y.C.); imasoel@ppukm.ukm.edu.my (I.-N.S.); azlina@ppukm.ukm.edu.my (N.M.); norliza_ssp@ppukm.ukm.edu.my (N.M.); 2Department of Anatomy, Faculty of Medicine, Universiti Kebangsaan Malaysia, Jalan Yaacob Latif, Bandar Tun Razak, Cheras 56000, Malaysia; apai.kie@gmail.com; 3Drug and Herbal Research Centre, Faculty of Pharmacy, Universiti Kebangsaan Malaysia Kuala Lumpur Campus, Jalan Raja Muda Abdul Aziz, Kuala Lumpur 50300, Malaysia; pyng@ukm.edu.my; 4Centre for Community Health Studies, Faculty of Health Science, Universiti Kebangsaan Malaysia Kuala Lumpur Campus Jalan Raja Muda Abdul Aziz, Kuala Lumpur 50300, Malaysia; ainijamil@ukm.edu.my; 5Department of Family Medicine, Faculty of Medicine, Universiti Kebangsaan Malaysia, Jalan Yaacob Latif, Bandar Tun Razak, Cheras 56000, Malaysia; azah@ppukm.ukm.edu.my

**Keywords:** bone mineral density, calcium, exercise, osteoporosis, osteopenia

## Abstract

*Background:* The current osteoporosis screening instruments are not optimized to be used among the Malaysian population. This study aimed to develop an osteoporosis screening algorithm based on risk factors for Malaysians. *Methods:* Malaysians aged ≥50 years (n = 607) from Klang Valley, Malaysia were interviewed and their bone health status was assessed using a dual-energy X-ray absorptiometry device. The algorithm was constructed based on osteoporosis risk factors using multivariate logistic regression and its performance was assessed using receiver operating characteristics analysis. *Results:* Increased age, reduced body weight and being less physically active significantly predicted osteoporosis in men, while in women, increased age, lower body weight and low-income status significantly predicted osteoporosis. These factors were included in the final algorithm and the optimal cut-offs to identify subjects with osteoporosis was 0.00120 for men [sensitivity 73.3% (95% confidence interval (CI) = 54.1%–87.7%), specificity 67.8% (95% CI = 62.7%–85.5%), area under curve (AUC) 0.705 (95% CI = 0.608–0.803), *p* < 0.001] and 0.161 for women [sensitivity 75.4% (95% CI = 61.9%–73.3%), specificity 74.5% (95% CI = 68.5%–79.8%), AUC 0.749 (95% CI = 0.679–0.820), *p* < 0.001]. *Conclusion:* The new algorithm performed satisfactorily in identifying the risk of osteoporosis among the Malaysian population ≥50 years. Further validation studies are required before applying this algorithm for screening of osteoporosis in public.

## 1. Introduction

Osteoporosis is a metabolic bone disease characterized by deterioration of bone microarchitecture and bone mass which reduces its strength and eventually increases the risk of fractures [1]. The prevalence of osteoporosis is expanding in line with the increase in the elderly population. Specifically, the Malaysian population aged ≥50 years is expected to increase from 5.3 million in 2013 to 13.9 million in 2050 [2]. Another forecast reported that Malaysia would experience a 3.55-fold increase in hip fracture incidence by 2050 compared to 2018 [3].

Clinically, osteoporosis is diagnosed based on bone mineral density (BMD) determined by dual-energy X-ray absorptiometry (DXA) [4]. A BMD value ≤ –2.5 standard deviations (SD) from the young adult mean (or a T score ≤ –2.5) indicates osteoporosis while a T-score between ≤ –1.0 and > –2.5 indicates osteopenia [4]. Although this classification system is simple, the accessibility of DXA in developing countries is limited due to the high cost of scanning and the low number of DXA machines [2]. Prioritizing individuals at risk for osteoporosis for DXA scan can reduce the competition for health resources. In Malaysia, DXA machines in public hospitals are reserved to confirm the bone health of patients with strong risk factors of osteoporosis but not for bone health screening [2]. 

An effective osteoporosis screening tool will be able to reduce the burden of DXA by prioritizing patients at high risk of osteoporosis for DXA scan. Several algorithms based on risk factors of osteoporosis have been used clinically, including Osteoporosis Self-Assessment Tool (OST) [5] and Osteoporosis Risk Assessment Tool (ORAI) [6]. These algorithms were developed using the data from Western populations, thus may not be suitable to be used among Asian populations due to the differences in genetics, lifestyle and environmental factors. Osteoporosis Self-Assessment Tool for Asians (OSTA) is an alternative screening algorithm developed for Asians [5,7]. However, our previous study showed that OSTA at its original cut-off (< –4 for high risk; –4 to –1 for low risk) values did not predict osteoporosis among Malaysian men and women >40 years [8]. 

Some studies reported that algorithms developed locally work better than OSTA in identifying subjects with osteoporosis [9,10]. In Malaysia, Lim et al. (2011) developed the Malaysian Osteoporosis Screening Tool (MOST), which calculates the risk of low BMD among midlife women based on age, years since menopause, body mass index and hip circumference (n = 514, mean age = 51.3 ± 5.4 years). MOST performed well among women surveyed (cut-off value ≥ 4, sensitivity = 80.2%, specificity = 55.5%) but it cannot be used in men [11]. Therefore, there is a paucity of local screening algorithm suitable to be used by both Malaysian men and women. 

The present study aims to develop an osteoporosis screening algorithm for the Malaysian population aged ≥50 years. We focused on the population residing in Klang Valley, Malaysia because it is a highly urbanized area. The Asian Osteoporosis Study identified that the incidence of hip fracture was higher in urbanized cities in Asia [12]. Since the risk factors of osteoporosis differ between sexes, separate algorithms were generated for men and women.

## 2. Materials and Methods 

### 2.1. Study Population

This study is a part of the cohort study investigating the bone health of Malaysians living in Klang Valley, Malaysia [8,13,14,15]. This cross-sectional study initially recruited 786 community-living Malaysians aged ≥40 years living in Klang Valley, Malaysia using quota sampling technique. Subjects were stratified based on sex (ratio men to women, 1:1) and ethnicity (Malays 45%, Chinese 45%, Indians and other ethnicities 10%) corresponding to the population demography of Klang Valley, Malaysia [16]. Invitations with specific inclusion and exclusion criteria were sent to community centers in Klang Valley and advertised in local newspapers and radio stations. Only subjects fulfilling the criteria in Table 1 were recruited. Informed consent was obtained from all the subjects prior to their enrolment. The study was approved by the Research Ethics Committee of Universiti Kebangsaan Malaysia (Approval code: UKM PPI/111/8/JEP-2017-761; Date: 27 December 2017). For this study, only subjects aged ≥50 years (n = 607) were included in the final analysis.

All subjects answered a questionnaire regarding their demographic details (ethnicity, occupation, education level, parity, current menstrual status, age of menarche and menopause), medical history, supplement intake, intake of dairy products (milk, cheese and yoghurt), coffee, tea and alcohol, and lifestyle practices (smoking habits and physical activity). Subjects were classified into the bottom 40% (B40, with monthly income < RM7,640 or USD 1868), the middle 40% (M40, with monthly income RM 7640-RM 15,159 or USD 1868–USD 3,707) and top 20% (T20, with monthly income RM ≥ 15,160 or USD ≥ 3707) household income groups [17]. For beverages, individuals with an intake of less than one unit per week were defined as non-drinkers. One unit of milk was defined as 200 mL, while one unit of coffee/tea was defined as one standard tea cup [18]. Those who consumed beverages or dairy products 3–4 days per week were defined as regular drinkers in the study. Alcohol unit was defined based on the recommendation by the National Health Service, UK [19]. Regular alcohol drinkers (3–4 days per week) or subjects who stopped drinking for the past 12 months were combined as ‘ever drinkers’. Meanwhile, those who rarely (one or fewer days per month) or never consumed were combined as ‘non-drinkers’. For smoking status, former smokers (abstained from smoking within the past 12 months) and current smokers were grouped as ‘ever smokers’. Subjects who consumed at least one tablet of calcium supplement for 3–4 days per week were considered as regular consumers.

The International Physical Activity Questionnaire (IPAQ) was used to assess the physical activity status of subjects [20]. Subjects were required to recall the average amount of time spent and frequency of carrying out vigorous and moderate physical activities as well as walking in a week. Subjects were categorized into inactive, minimally-active or HEPA (health-enhancing physical activity) active based on total metabolic equivalents (MET) score or other additional criteria [18]. For analysis in this study, both the HEPA active and minimally-active groups were combined as ‘active’ group.

### 2.2. Anthropometric and Bone Mineral Density (BMD) Measurement

A stadiometer (SECA, Hamburg, Germany) was used to measure subjects’ standing height without shoes to the nearest 1 cm. A body composition analyzer (TANITA, Tokyo, Japan) was used to determine their body weight with light clothing but without shoes to the nearest 0.1 kg. Body mass index (BMI: kg/m^2^) was calculated by dividing body weight (kg) by the square of body height (m^2^). BMI was classified as underweight (<18.5 kg/m^2^), normal (18.5–24.9 kg/m^2^) and overweight (>24.9 kg/m^2^) for those aged <65 years [21]. Meanwhile, for subjects aged >65 years, a BMI <22 kg/m^2^ was classified as underweight, 22–27 kg/m^2^ as normal and >27 kg/m^2^ as overweight [22]. The waist circumference of subjects was measured while they maintained their standing position [23].

A DXA device (Hologic Discovery QDR Wi densitometer, Hologic, MA, USA) determined subjects’ BMD at the lumbar spine (anteroposterior, L1-L4) and left hip. The BMD values (in g/cm^2^) were converted to T-score based on the reference values of the Singaporean population according to ethnicity. The World Health Organization (WHO) criteria were used to classify the bone health status of subjects into osteoporosis (T-score ≤ –2.5), osteopenia (T-score ≤ –1 and > –2.5) and normal (T-score > –1.0). The DXA measurement was performed by an experienced technologist. Daily calibration was conducted using a phantom. The short-term in vivo coefficient of variation for this device was 1.8% and 1.2% for lumbar spine and total hip, respectively.

### 2.3. Statistical Analysis

Normality of the data was determined using the Kolmogorov–Smirnov test. Logistic regression analysis was used to determine the risk factors of osteoporosis and to develop the new algorithm. All continuous variables were entered into the equation using the stepwise method. All categorical variables were force-entered in the second step. The effect size of each risk factor was expressed as odds ratio (OR) and 95% confidence interval (CI). The variables involved in the development of the final algorithm were determined via logistic regression analysis. The logistic regression model was used as the basis in formulating the screening algorithm of this study Equation (1), whereby P = expected probability; *β* = coefficient of covariates; *X* = continuous or categorical covariate [24]. Outlier cases with standardized residual values >3 were removed from the regression model. The performance of the algorithm in identifying subjects with osteoporosis with reference to DXA was assessed using receiver operating characteristic (ROC) curve analysis. The sensitivity, specificity and AUC for each index were determined using ROC. An AUC value of less than 0.5 indicates a poor performance in predicting individuals at risk of osteoporosis. An AUC value between 0.7–0.8 is considered as acceptable performance, 0.8–0.9 as excellent performance and above 0.9 is considered as outstanding performance [25]. The optimal cut-off values were obtained by coordinate tracing of the ROC curve. The coordinate values that gave the best Youden’s index (J = sensitivity + specificity − 1) were selected as the best cut-off values [26,27]. Statistical Package for Social Science version 23.0 (IBM, Armonk, USA) was used for all statistical analyses in the study. A p-value < 0.05 was considered to be statistically significant.
(1)$$P=\frac{1}{1+e^{-\left(\beta 0+\beta 1X1+\beta 2X2+\ldots \beta nXn\right)}}$$

## 3. Results

Data of 607 subjects were included in the final analysis, of which 303 (49.9%) were men and 304 were (50.1%) women. The mean age of men and women were 61.98 ± 6.78 and 59.73 ± 6.51 years, respectively. The subjects consisted of Chinese (50.1%), Malays (39.2%) and Indians or other ethnic groups (10.7%). A majority of the subjects were not habitual consumers of dairy products (64.1%) or calcium supplements (83.7%). Besides, 54.2% of the subjects were actively involved in physical activity. The prevalence of osteoporosis among men and women was 9.9% and 20.1%. In addition, 33.3% of men and 49.0% of women were found to have osteopenia (Table 2).

### 3.1. Model Development

Logistic regression results showed that increased age (OR = 1.282, 95% CI: 1.088–1.510, *p* = 0.003) predicted greater risk of osteoporosis in men. Increased body weight (OR = 0.711, 95% CI: 0.599–0.844, *p* < 0.001) and being physically active (vs. inactive, OR = 0.007, 95% CI: <0.001–0.105, *p* < 0.001) predicted lower risk of osteoporosis in men. In women, increased age (OR = 1.124, 95% CI: 1.064–1.187, *p* < 0.001) and B40 income status (vs. M40, OR = 14.978, 95% CI: 3.643–61.577, *p* < 0.001) predicted higher risk of osteoporosis. Increased body weight (OR = 0.875, 95% CI: 0.836–0.917) was protective against osteoporosis in women (*p* < 0.001) (Table 3).

The predictors of osteoporosis aforementioned with their corresponding unstandardized regression coefficient values (Table 3) were inserted into the formula to form the algorithm for men Equation (2) and women Equation (3).
(2)$$P=\frac{1}{1+e^{-\left[3.009+0.249\left(age\;in\;years\right)-0.341\left(body\;weight\;in\;kg\right)-4.93\left(physical\;activity\;status\right)\right]}}$$

Notes: Physical activity status, active = ‘1′ and inactive = ‘0′.
(3)$$T=\frac{1}{1+e^{-\left[-1.425+0.117\left(age\;in\;years\right)-0.133\left(weight\;in\;kg\right)+2.707\left(monthly\;income\;category\right)\right]}}$$

PNotes: Income status, low income (B40) = ‘1′ or high income (M40 and T20) = ‘0′.

### 3.2. Diagnostic Performance of Algorithms and Comparison with OSTA

The predictive performance of the new algorithms was determined using ROC analysis. Both the algorithms demonstrated significant AUC values in predicting osteoporosis among the subjects (*p* < 0.001). At cut-off ≥ 0.00120, the algorithm for men showed a satisfactory performance with a sensitivity of 73.3% (95% CI = 54.1%–87.7%), specificity of 67.8% (95% CI = 62.7%–85.5%) and AUC of 0.705 (95% CI = 0.608–0.803). At cut-off ≥ 0.161, the algorithm for women showed a satisfactory performance with a sensitivity, specificity and AUC of 75.4% (95% CI = 61.9–73.3), 74.5% (95% CI = 68.5%–79.8%) and 0.749 (95% CI = 0.679–0.820), respectively (Table 4). 

## 4. Discussion

This study has developed two different osteoporosis screening algorithms based on sex because the risk factors were different between men and women. The algorithm for men involved age, body weight and physical activity status. Increased age and low body weight are the most common risk factors contributing to osteoporosis. Hence, age and body weight were included in some of the established screening algorithms, such as Osteoporosis Risk Assessment Instrument (ORAI) [6] and OSTA [5]. Physical activity status was also included as one of the negative predictors of osteoporosis in the screening algorithm for men. Weight-bearing physical activities like jogging or climbing stairs could exert mechanical loading on the bone and stimulate bone mass accrual to accommodate the load. Therefore, physical activities could reduce bone loss, increase bone strength and prevent osteoporosis in the elderly population [28]. Thus, it is important to consider physical activity status in the screening algorithm. Physical activity status of this study was assessed using IPAQ. For quick and convenient self-evaluation in the future, a modified assessment of physical activity status may be necessary so that the public can input their status into the equation easily.

The osteoporosis screening algorithm developed for women includes age, body weight and income status as predictors. Similar to men, increased age and low body weight were significant positive predictors of osteoporosis in women. Income status is rarely considered by any established screening algorithm because it varies among countries. In this study, low monthly income is a positive predictor of osteoporosis among women. The low socioeconomic status may prevent women from participating in exercise or obtaining a balanced diet [29]. The lack of disposable income will also hinder women’s access to calcium and vitamin D supplements, which is shown to prevent fractures related to osteoporosis [30,31]. Estrogen deficiency after menopause will accelerate bone resorption rate and reduce bone formation, leading to osteoporosis [32]. Menopause is the primary reason for bone loss in women and primary osteoporosis occurs 10–15 years after menopause [32]. Most of the osteoporosis screening algorithms available currently include “years since menopause” as one of the variables, which measures the duration of estrogen deprivation in women. For instance, a screening algorithm developed by Hawker et al. (2012) to identify Canadian women (age ≥40 years) with low bone mass includes “years since menopause” as a predictor [33]. “Years since menopause” was considered but it was not predictive of bone health status among women in this study. 

Generally, the osteoporosis screening algorithms developed for both sexes in the study showed a convincing performance with regards to the sensitivity, specificity and AUC. Moreover, these algorithms demonstrated a better performance in identifying subjects with osteoporosis as compared to using OSTA with its existing cut-off values in the same cohort of subjects [16]. OSTA was not predictive among these subjects [8], as evidenced by the low sensitivity, specificity and AUC values (Table 5), which are in contrast to the findings by Koh et al. (2001) [5]. This discrepancy is due to the differences in characteristics of the cohort for the establishment of OSTA and the Malaysian populations. The study of Koh et al. (2001) only involved postmenopausal women and the majority of the subjects originated from East Asia. The present study involved both sexes and three major ethnic groups in Malaysia. In addition, the subjects recruited by Koh et al. (2001) were from the clinics and tertiary centers, while the subjects of the current study were recruited from the community-dwelling population. The adjusted OSTA cut-off values (1.8 for men, 0.8 for women) were proven to be effective in identifying individuals with osteoporosis within the same cohort of subjects previously [8]. The performance of the newly developed algorithms is comparable with the performance of OSTA at its adjusted cut-off values. The overall sensitivity, specificity, and AUC of the new algorithm were on par with OSTA with adjusted cut-off values (Table 5). 

Several country-specific osteoporosis screening algorithms developed are effective in identifying individuals at risk of osteoporosis. For example, Beijing Friendship Hospital Osteoporosis Self-Assessment Tool for Elderly Male (BFH-OSTM) comprising two predictors, body weight and history of fracture, was developed by Lin et al. (2017) among 1870 Han Chinese men (age: ≥50 years) [34]. At the cut-off of 70, BFH-OSTM was effective in identifying men with osteoporosis (sensitivity = 85%, specificity = 53%, AUC = 0.763). In the subsequent validation phase (n = 574 men), the performance of BFH-OSTM (sensitivity = 79.83%, specificity = 62.20%, AUC = 0.795) was superior to OSTA (sensitivity = 50.42%, specificity = 82.20%, AUC = 0.732) [34]. Oh et al. (2013) developed the Korean Osteoporosis Risk-Assessment Model (KORAM) among postmenopausal women (development phase: n = 1209; validation phase: n = 1046) with mean age of 63.5 ± 8.9 year. KORAM included age, body weight and hormone replacement therapy as predictors. KORAM (cut-off <s–9, sensitivity = 84.8%–91.2%, specificity = 50.6%–51.6%, AUC: 0.682–0.709) showed a better performance compared to OSTA (cut-off <0, sensitivity = 94.2%–96.8%, specificity = 28.3%–29.2%, AUC: 0.617–0.626) in predicting osteoporosis [34]. The performance of our algorithms for the Malaysian population was on par with the aforementioned country-specific osteoporosis screening algorithms. 

Several limitations should be considered in the present study. Firstly, the algorithm developed was not validated in another cohort due to the constraint of resources. Secondly, the algorithm is only suitable to be used among those without apparent risk of osteoporosis, since they are not usually referred to DXA scan. Thirdly, since the subjects involved in this study comprised only the three major ethnic groups in Malaysia, it may not apply to ethnic minorities in Malaysia or outside of Malaysia. Thus, generalization of the observations in this study must be performed with caution. Nevertheless, these algorithms could potentially help to identify individuals at risk of osteoporosis and prioritize them for BMD assessment. 

## 5. Conclusions

In conclusion, the newly developed screening algorithms developed based on risk factors of osteoporosis can identify individuals at risk for osteoporosis. They also perform better than OSTA used widely in the screening of osteoporosis. The new algorithms can be used clinically, pending validation in a larger cohort, to reduce the burden of the DXA device and prevent unnecessary DXA scans. 

## Figures and Tables

**Table 1 ijerph-17-02526-t001:** Inclusion and exclusion criteria used in this study.

Inclusion Criteria	Exclusion Criteria
Malaysians	Diagnosed with bone diseases (Paget’s disease, osteogenesis imperfect, osteomalacia, rickets).
Residing in Klang Valley, Malaysia	Diagnosed with conditions that alter bone metabolism (hypo/hypercalcemia, hypo/hyperthyroidism, hypo/hypergonadism).
No apparent risk of osteoporosis	Receiving therapeutic agents (thiazide diuretics, glucocorticoids, thyroid supplements, anticonvulsants, antidepressants and osteoporosis treatment agents etc.) that alter bone metabolism.
	Having mobility problems, requiring a walking aid, fractured six months prior to the screening date, having metal implants at the site of scan.
	Suffered a low impact fracture after the age of 50 years.

**Table 2 ijerph-17-02526-t002:** Characteristics of the study population.

Variable of Interest	Categories	Men (n = 303)		Women (n = 304)		Overall (n = 607)	
		Mean (SD)	Median (IQR)	Mean (SD)	Median (IQR)	Mean (SD)	Median (IQR)
**Age (years)**		61.98 (6.78)	62.00 (10.0)	59.73 (6.51)	59.00 (9.00)	60.85 (6.74)	60.00 (9.00)
Age of menarche (years)		-	-	13.27 (1.85)	13.00 (2.00)	13.27 (1.85)	13.00 (2.00)
Age of menopause (years)		-	-	44.01 (18.17)	51.00 (5.00)	44.01 (18.17)	51.00 (5.00)
Years since menopause (years)		-	-	8.41 (7.5)	7.00 (11.00)	8.41 (7.5)	7.00 (11.00)
**Body Anthropometry**	Weight (kg)	69.74 (9.97)	68.90 (13.1)	60.25 (11.46)	59.50 (13.20)	64.99 (11.74)	64.30 (15.40)
	Height (m)	166.02 (10.35)	166.50 (7.80)	154.21 (5.73)	153.90 (7.30)	160.10 (10.23)	160.80 (13.10)
	BMI (kg/m^2^)	25.14 (3.46)	24.80 (4.30)	25.34 (4.71)	24.75 (6.0)	25.24 (4.13)	24.80 (4.90)
	Waist circumference (cm)	88.95 (10.76)	88.60 (13.00)	83.24 (11.40)	83.00 (13.00)	86.09 (11.44)	86.00 (12.00)
**Bone Mineral Density**	Lumbar spine (g/cm^2^)	0.99 (0.19)	1.00 (0.20)	0.87 (0.15)	0.87 (0.20)	0.93 (0.18)	0.92 (0.22)
	Left hip (g/cm^2^)	0.91 (0.13)	0.91 (0.18)	0.81 (0.12)	0.81 (0.15)	0.86 (0.14)	0.86 (0.17)
**Demography and BMI Status**		**number (percentage; %)**					
Age Range (years)	50–59	116 (38.3)		162 (53.3)		278 (45.8)	
	60–69	138 (45.5)		114 (37.5)		252 (41.5)	
	>70	49 (16.2)		28 (9.2)		77 (12.7)	
Ethnics	Malay	114 (37.6)		124 (40.8)		239 (39.2)	
	Chinese	156 (51.5)		148 (48.7)		304 (50.1)	
	Indians/others	33 (10.9)		21 (10.5)		65 (10.7)	
Marital Status	Single	7 (2.3)		25 (8.2)		32 (5.3)	
	Married	296 (97.7)		279 (91.8)		575 (94.7)	
Nature of Occupation	Manual	13 (4.3)		10 (3.3)		23 (3.8)	
	Sedentary	290 (95.7)		294 (96.7)		584 (96.2)	
Estimated Monthly Income	B40 (<RM 7640)	278 (91.7)		292 (96.1)		570 (93.9)	
	M40 (RM 7640–RM 15,159)	25 (8.3)		12 (3.9)		37 (6.1)	
Highest Education Level	No formal education and Primary school	29 (9.6)		30 (9.9)		59 (9.7)	
	Secondary school	130 (42.9)		165 (54.3)		295 (48.6)	
	Certificate/diploma	66 (21.8)		56 (18.4)		122 (20.1)	
	University degree or above	78 (25.7)		53 (17.4)		131 (21.6)	
Parity	Nulliparous	-		49 (16.1)		49 (16.1)	
	1–3 Pregnancies	-		131 (43.1)		131 (43.1)	
	More than 3 Pregnancies	-		124 (40.8)		124 (40.8)	
	Underweight (<18.5 kg/m^2^)	25 (8.3)		34 (11.2)		59 (9.7)	
BMI Classification	Normal (18.5–24.9 kg/m^2^)	145 (47.9)		135 (44.4)		280 (46.1)	
	Overweight (>25 kg/m^2^)	133 (43.9)		135 (44.4)		268 (44.2)	
**Lifestyle**		**number (percentage; %)**					
Regular Dairy Product Intake	Yes	80 (26.4)		138 (45.4)		218 (35.9)	
	No	223 (73.6)		166 (54.6)		389 (64.1)	
Regular Calcium Supplement Users	Yes	34 (11.2)		65 (21.4)		99 (16.3)	
	No	269 (88.8)		239 (78.6)		508 (83.7)	
Regular Coffee/Tea Consumption	Yes	38 (12.5)		76 (25.0)		114 (18.8)	
	No	265 (87.5)		228 (75.0)		493 (81.2)	
Regular Alcohol Intake	Yes	66 (21.8)		14 (4.6)		80 (13.2)	
	No	237 (78.2)		290 (95.4)		527 (86.8)	
Smoking	Yes	126 (41.6)		6 (2.0)		132 (21.7)	
	No	177 (58.4)		298 (98.0)		475 (78.3)	
Physical Activity	Inactive (<600 MET/min)	126 (41.6)		152 (50.0)		278 (45.8)	
	Active (>600 MET/min)	177 (58.4)		152 (50.0)		329 (54.2)	
**Fracture History & Bone Health Status**		**number (percentage; %)**					
History of Fracture	Yes	18 (5.9)		14 (4.6)		32 (5.3)	
	No	285 (94.1)		290 (95.4)		575 (94.7)	
Osteoporosis Self-Assessment Tool for Asians	Low risk	249 (82.2)		202 (66.4)		451 (74.3)	
	Moderate risk	52 (17.2)		81 (26.6)		133 (21.9)	
	High risk	2 (0.7)		21 (6.9)		23 (3.8)	
Bone Health Status	Normal (T-score > –1.0)	172 (56.8)		94 (30.9)		266 (43.8)	
	Osteopenia (T-score ≤–1 and > –2.5)	101 (33.3)		149 (49.0)		250 (41.2)	
	Osteoporosis (T-score ≤–2.5)	30 (9.9)		61 (20.1)		91 (15.0)	

B40, bottom 40; BMI, body mass index; IQR, interquartile range; MET, metabolic equivalents; M40, middle 40; min, minute; SD, standard deviation.

**Table 3 ijerph-17-02526-t003:** Risk factors of osteoporosis.

Variables	Odds Ratio (OR)	95% CI for OR		B	*p*-Value
		Lower	Upper		
**Men**					
**Age**	1.282	1.088	1.510	0.249	0.003
**Body weight**	0.711	0.599	0.844	–0.341	<0.001
**Physical activity**	0.007	<0.001	0.105	–4.93	<0.001
*Active vs Inactive (ref.)*					
**Constant of the model**	-	-	-	3.009	0.593
**Women**					
**Age**	1.124	1.064	1.187	0.117	<0.001
**Body weight**	0.875	0.836	0.917	–0.133	<0.001
**Monthly income**	14.978	3.643	61.577	2.707	<0.001
*B40 vs. M40 (ref.)*					
**Constant of the model**	-	-	-	–1.427	0.495

The bolded *p*-values are statistically significant. Cases with standardized residual >3 are removed; thus, the final cases retained in the multivariate logistic regression model are 287 men and 297 women. Notes: “Active” group constituted moderately active and HEPA-active groups as determined by IPAQ. Abbreviation: B, unstandardized regression coefficient; CI, confidence interval; OR, odds ratio; vs., versus.

**Table 4 ijerph-17-02526-t004:** The performance of the new osteoporosis screening algorithms.

Sex	Cut-off Value	Sensitivity (%)	95% CI	Specificity (%)	95% CI	J	AUC	95% CI	*p*-Value
**Men**	≥0.00120	73.3	54.1–87.7	67.8	62.7–85.5	0.411	0.705	0.608–0.803	<0.001
**Women**	≥0.161	75.4	61.9–73.3	74.5	68.5–79.8	0.499	0.749	0.679–0.820	<0.001

The bolded *p*-values are statistically significant. Abbreviation: AUC, area under curve; CI, confidence interval; J, Youden’s index.

**Table 5 ijerph-17-02526-t005:** Comparison of the performance of the new osteoporosis screening algorithms with Osteoporosis Self-Assessment Tool for Asians (OSTA).

Screening Tool	Cut-off Value	Sensitivity (%)	Specificity (%)	AUC	95% CI	*p*-Value
**Men**						
New algorithm	≥0.00120	73.3	67.8	0.705	0.608–0.803	<0.001
OSTA (original cut-off) [8]	< –4	0	99.4	0.497	0.393–0.601	0.957
OSTA (modified cut-off) [8]	≤1.8	81.3	61.4	0.699	0.610–0.787	<0.001
**Women**						
New algorithm	≥0.161	75.4	67.8	0.749	0.679–0.820	<0.001
OSTA (original cut-off) [8]	< –4	20.3	97.6	0.587	0.504–0.669	0.027
OSTA (modified cut-off) [8]	≤0.8	81.5	55.5	0.679	0.612–0.745	<0.001

The bolded *p*-values are statistically significant. Abbreviation: AUC, area under curve; CI, confidence interval.

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
