# Peer review of "Development of Osteoporosis Screening Algorithm for Population Aged 50 Years and above in Klang Valley, Malaysia"

_ijerph, 2020, doi:10.3390/ijerph17072526_

Round 1
Reviewer 1 Report
The study focuses on the current issue of screening for osteoporosis in the Malaysian population. The manuscript is concise and clearly. As the results of the study have evident potential for practical application, it will be of interest to the audience of the journal. However, some methodological aspects of the study are of concern. Therefore, I believe that the article should undergo a major revision.
- The study included individuals aged 40 or older. According to the WHO criteria, for men and women under the age of 50, Z-score is recommended for BMD assessment. Meantime, T-score should be applied for men and women >50 years old and postmenopausal women. According to the described methods, the entire population of this study was evaluated with T-score. It is better to exclude people under 50 years old and women under 50 with preserved menstrual function, or reevaluate them with Z-score.
- Line 131-132. How was the short-term coefficient of variation determined in this case, and what is its acceptable cut-off value? Were the values >1.5% considered acceptable?
- Since two different versions of the screening algorithm were generated for men and women, the characteristics of these groups should be given separately.
- What was the prevalence of osteoporosis and osteopenia in men and women?
- Table 2. As not all variables could be distributed normally, it is better to present the data as medians and 25, 75 percentiles.
- Table 2. For readability, it is recommended to split the table into subsections.
- Line 96-99. The definition of income is not entirely clear.
Does this mean monthly or annual income?
Author Response
Dear Reviewer,
Thank you for your meticulous review. We value your constructive comments and have replied to each of the comment in the attached response sheet. All changes in the text are tracked. We hope the revised manuscript could achieve your expectations.
Thank you.

Reviewer 2 Report
Please see the attached pdf file for my comments and recommended revisions. Overall, well done!

Author Response

(The authors gave the same response as above.)

Round 2
Reviewer 1 Report
The article has undergone substantial revision. The comments made during the reviewing were taken into account. I could recommend the manuscript for publication.